# Semi-supervised Context-aware Multi-Organ Segmentation

Jiacheng Wang[1] and Yongkang Ma[1]

Manteia Technologies Co.,Ltd, Xiamen, China
`{wangjiacheng, yongkangma}@manteiatech.com`

**Abstract.** The automatic segmentation of organs at risk is extremely dedicated to the clinical assistance that can significantly reduce the clinical resource cost. However, training such a good enough model usually requires a large amount of labeled data, or the model performance is likely to meet heavy drop. Semi-supervised training strategies are proved to be an effective solution to reduce the reliance of labeled data. In this paper, we develop a powerful semi-supervised learning framework to address the label-efficient multi-organ segmentation. The experiments are conducted on the MICCAI FLARE 2022 dataset, where the results show that the semi-supervised learning strategy has significant performance boost.

**Keywords:** Semi-supervised, Multi-Organ segmentation

## 1    Introduction

The automatic segmentation of organs at risk [5,6,12,10,25,14,13,22] is extremely dedicated to the clinical assistance that can significantly reduce the clinical resource cost. However, training such a good enough model usually requires a large amount of labeled data, or the model performance is likely to meet a heavy drop. Particularly for the clinics, it is well-known that the dense-level annotation needs expertise knowledge to make the precise decision. Thus, simply envolving a large group of annotators to handle the annotation task is not helpful. To develop a label-efficient model seems to be the only one alternation that is also desperately needed nowadays.

There are plenty of wise solutions to reduce the reliance of labeled data, such as weakly supervised learning, self-supervised learning during model pretraining [24,21,11,2,1,20], and the most popular one of which is the semi-supervised learning methods [19,7,23,4,18]. As a famous research direction, a lot of studies have been conducted in distinct areas and imaging modalities, such as natural images, radiation images, fundus iamges, and so on. No matter what the imaging is, these methods have similar underlying principles that are to produce convincing pseudo labels for semi-supervision. In this group, the most simple and straightforward tool is to produce pseudo label using trained models and then using a post-processing strategy to reduce the false positives. Here, in this work, we follow this group and apply it into the 3D context-aware organ segmentation models.

## 2    Method

### 2.1    Preprocessing

The preprocessing steps include 1) reorienting images to the left-posterior-inferior (LPI) view by flipping and reordering, 2) resampling images to keep a fixed size $(160, 160, 160)$, and 3) normalizing images to reduce the distribution error by z-score normalization.

### 2.2    Proposed Method

Our model follows the standard 3D U-Net design to achieve the coarse-to-fine organ segmentation. It consists of two stages, where the first stage is to coarsely localize the organs and the second stage is to precisely segment the boundaries.

Specifically speaking, the two stages share same network architecture design as Fig. 1 shows, which is just a standard 3D U-Net. They differ in the inputting data where the fist stage takes the original data as input while the second stage takes the cropped prediction of the first stage as input.

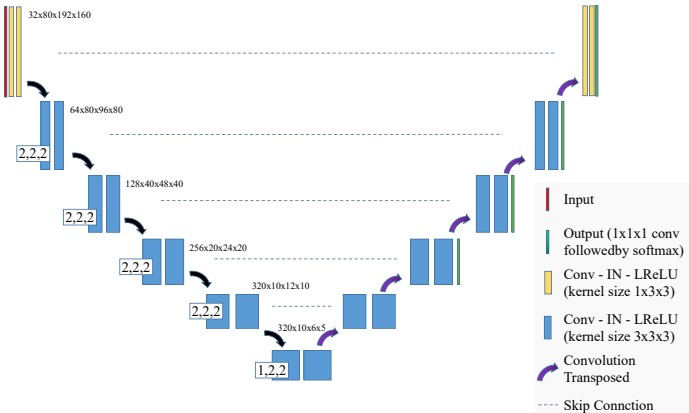

**Fig. 1.** The used network architecture of one stage.

### 2.3    Unlabel Data

To use the unlabeled volumes, we first train a good segmentation model under full supervision using the labeled data. Then, the model calculates the prediction of each unlabeled volume to obtain the pseudo label. As the existing false positives heavily affect the label quality, we apply the post-processing strategy like connective field analysis to improve the predictions.0 After generating the pseudo labels, we merge these data into the label data and re-train the segmentation models from scratch.

### 2.4   Objectives

We use the summation between Dice loss and cross entropy loss because compound loss functions have been proved to be robust in various medical image segmentation tasks [15].

## 3   Experiments

### 3.1   Dataset and evaluation measures

The FLARE2022 dataset is curated from more than 20 medical groups under the license permission, including MSD [17], KiTS [8,9], AbdomenCT-1K [16], and TCIA [3]. The training set includes 50 labelled CT scans with pancreas disease and 2000 unlabelled CT scans with liver, kidney, spleen, or pancreas diseases. The validation set includes 50 CT scans with liver, kidney, spleen, or pancreas diseases. The testing set includes 200 CT scans where 100 cases has liver, kidney, spleen, or pancreas diseases and the other 100 cases has uterine corpus endometrial, urothelial bladder, stomach, sarcomas, or ovarian diseases. All the CT scans only have image information and the center information is not available.

The evaluation measures consist of two accuracy measures: Dice Similarity Coefficient (DSC) and Normalized Surface Dice (NSD), and three running efficiency measures: running time, area under GPU memory-time curve, and area under CPU utilization-time curve. All measures will be used to compute the ranking. Moreover, the GPU memory consumption has a 2 GB tolerance.

### 3.2   Implementation details

**Environment settings**  The detailed environment settings have been shown in Table 1 where we use a single Titan 3090 GPU with 24 GB memory to train the entire network.

**Table 1.** Development environments and requirements.

| | |
|---|---|
| Windows/Ubuntu version | Ubuntu 16.04.5 LTS |
| CPU | Intel(R) Core(TM) i9-7700X CPU@3.30GHz |
| RAM | 16×4GB; 2.67MT/s |
| GPU (number and type) | One NVIDIA 3090 24G |
| CUDA version | 11.0 |
| Programming language | Python 3.9 |
| Deep learning framework | Pytorch (Torch 1.10, torchvision 0.2.2) |
| Specific dependencies | |
| (Optional) Link to code | |

**Training protocols** The details of training protocols have been shown in Table 2 and Table 3. We use the standard "he" normal initialization to initialize the model paramters with the consideration to avoid overfitting. The batch size is set to four considering the memory limitation and the computation efficiency. For the optimizer, we choose to utilize the AdamW optimizer emprically with an initial learning rate of 0.001. For stable training process, we apply the decay schedule to reduce the learning rate half each 200 epochs.

**Table 2.** Training protocols at the first stage.

| | |
|---|---|
| Network initialization | "he" normal initialization |
| Batch size | 4 |
| Patch size | 192×192×192 |
| Total epochs | 1000 |
| Optimizer | AdamW |
| Initial learning rate (lr) | 0.001 |
| Lr decay schedule | halved by 200 epochs |
| Training time | 72.5 hours |
| Number of model parameters | 41.22M[1] |
| Number of flops | 59.32G[2] |
| $CO_2$eq | 1 Kg[3] |

**Table 3.** Training protocols at the second stage.

| | |
|---|---|
| Network initialization | "he" normal initialization |
| Batch size | 4 |
| Patch size | 160×160×160 |
| Total epochs | 1000 |
| Optimizer | AdamW |
| Initial learning rate (lr) | 0.001 |
| Lr decay schedule | halved by 200 epochs |
| Training time | 72.5 hours |
| Number of model parameters | 41.22M[4] |
| Number of flops | 59.32G[5] |
| $CO_2$eq | 1 Kg[6] |

## 4    Results and discussion

### 4.1    Quantitative results on validation set

The averaged DSC scores of models trained under full-supervision and semi-supervision on the 10 cases validation set are respectively 0.8345 and 0.8523. It could be seen that the pseudo labeled data has performance boost for the segmentation modeling.

### 4.2    Qualitative results on validation set

This part is optional during validation phase since you do not have validation ground truth.

**Table 4.** Test results.

| Organs | DSC score |
|---|---|
| Liver | 0.9410 |
| RK | 0.9003 |
| Spleen | 0.9005 |
| Pancreas | 0.7585 |
| Aorta | 0.9352 |
| IVC | 0.8513 |
| RAG | 0.6579 |
| LAG | 0.7139 |
| Gallbladder | 0.6854 |
| Esophagus | 0.7522 |
| Stomach | 0.8021 |
| Duodenum | 0.6951 |
| LK | 0.8823 |
| Mean | 0.8058 |

### 4.3   Limitation and future work

The first problem of semi-supervised organ 3D segmentation is the memory limitation. As the previous tables have shown, the batch size is limited to 4 only which is a relatively small number. When performing the semi-supervised learning process, the data in one minibatch requires the strongly correct data to guide the optimization. With a limited and small mini-batch, the pesudo data will cover a large proportion so that the optimization will be affected and the final performance is poor.

## 5   Conclusion

In this work, we take the attempt to develop a semi-supervised learning framework to reduce the label reliance for multi-organ segmentation. We build a coarse-to-fine 3D organ segmentation model and train it under a supervised training manner. The model after training on the unlabeled data has a slight performance improvement. Due to the memory limitation, the performance has not been improved a lot, which can be further discussed in the future work.

**Acknowledgements** The authors of this paper declare that the segmentation method they implemented for participation in the FLARE 2022 challenge has not used any pre-trained models nor additional datasets other than those provided by the organizers. The proposed solution is fully automatic without any manual intervention.

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
