# OpenReview forum: "Semi-supervised Context-aware Multi-Organ Segmentation"
_MICCAI.org/2022/Challenge/FLARE_

### Official Review · Reviewer_mtFC · 2022-09-18
**No description about the efficiency, especially the inference time.**

**Rating:** 5
**Confidence:** 4

**Review:**

Strengths: The proposed method achieves the DSC of 0.8058 for multi-organ segmentation via two-stage coarse-to-fine framework.
Weaknesses:
There is no description about the efficiency of the method, especially the inference time. The two-stage framework is time-consuming.
The DSC still needs to be improved.

---

### Official Review · Reviewer_8SuU · 2022-09-19
**Coarse-to-fine U-Net for multi-organ segmentation**

**Rating:** 6
**Confidence:** 4

**Review:**

1. This work proposes a pseudo method for semi-supervised multi-organ segmentation. It first uses a 3D U-Net for coarse segmentation and then uses the same structure for refining the cropped prediction. The proposed method can use unlabeled data to achieve better performance.
2. The author describes the background and challenges of multi-organ segmentation but does not clearly explain the "Context-aware" in the title.
3. The author needs to double-check his/her manuscripts, e.g., section 4.2 is redundant.

---

### Official Review · Reviewer_CKoW · 2022-09-20
**The paper is well written but lacks novelty**

**Rating:** 4
**Confidence:** 5

**Review:**

The authors retrain the segmentation model using a vanilla pseudo-label strategy without other improvements, and the validation accuracy of the segmentation is not high.

---

### Official Review · Reviewer_rFKC · 2022-09-20
**Rough and incomplete**

**Rating:** 3
**Confidence:** 3

**Review:**

In this paper, the authors attempted to develop a semi-supervised learning framework to reduce the label reliance for multi-organ segmentation

Deficiencies:
1. The key methods should be described more clearly in the abstract.
2. The preprocessing methods should be described more clearly in subsection 2.1.
3. The descriptions in Figures 1 and subsection 2.1 do not match.
4. The authors should add strategies to improve inference speed and reduce resource consumption and a clear description of post-processing in section 2.
5. In Table 2 and Table 3, the authors should add the loss function in the next row after Training time.
6. The authors should visualize some good cases and bad cases, then analyze the reasons for the appearance of good cases and bad cases in section 4.
7. The authors should add segmentation efficiency results and segmentation efficiency analysis in section 4.
8. The author should make a detailed comparison between the results of using unlabeled data and not using unlabeled data in section 4.

---

### Meta-Review · Program_Chairs · 2022-09-28

**Recommendation:** Major Revision
**Confidence:** 5

**Metareview:**

Figure 1 Suspected plagiarism!!! Please re-draw it.

Reviewers raise many concerns and suggestions. Please address all comments in the revised manuscript.